# Beliefs and Use of Palpatory Findings in Osteopathic Clinical Practice: A Qualitative Descriptive Study among Italian Osteopaths

**DOI:** 10.3390/healthcare10091647

**Published:** 2022-08-29

**Authors:** Andrea Bergna, Matteo Galli, Francesco Todisco, Francesca Berti

**Affiliations:** 1Research Department, SOMA Istituto Osteopatia Milano, 20126 Milan, Italy; 2AISO-Associazione Italiana Scuole di Osteopatia, 65125 Pescara, Italy; 3Independent Researcher, 20100 Milan, Italy

**Keywords:** biopsychosocial approach, evidence-based practice, hands-on hands-off, osteopathic assessment, osteopathic clinical practice, osteopathic identity, osteopathy in Italy, palpatory finding, qualitative research, somatic dysfunction

## Abstract

The Italian government has started the regulatory process of osteopathy to include it among the healthcare professions mentioning terms, such as “perceptual palpation” and “somatic dysfunction” within the professional profile. ‘Palpatory findings’ are one of the multidimensional aspects that can inform osteopathic clinical reasoning. The non-regulated educational system has led to heterogenic professionals working in Italy, thus, the aim of this study was to investigate how Italian experts use palpatory findings in their clinical practice. A total of 12 experts were selected to participate in four virtual focus groups. A qualitative inductive approach with a constructivist paradigm was chosen to describe the results. The themes that emerged were: osteopathic identity; evaluation; osteopathic diagnosis; and sharing with different recipients. Participants agreed on the peculiarity and distinctiveness of osteopathic palpation, but there was some disagreement on the clinical significance of the findings, highlighting a complex multidimensional approach to diagnosis and treatment. The results seem to reflect the history of the profession in Italy, which has evolved quickly, leading professionals to seek new paradigms blending tradition and scientific evidence. The authors suggest further investigation to verify the state of art among osteopaths not involved in research or a broader consensus of the results.

## 1. Introduction

Osteopathy is a ‘person-centered’ approach to the prevention, diagnosis, and treatment of illness and injury. Osteopaths are primary-contact health-care professionals that use a range of techniques, including ‘hands-on’ manual techniques for assessment and diagnosis to treat various health conditions, as well as musculoskeletal structural problems that influence the body’s physiology.

Healthcare recognition of the osteopathic profession is being implemented in Italy, where this profession enjoys growing popularity. Indeed, according to estimates from an Italian population survey carried out in 2017, one in five Italians, or approximately 10 million citizens, have been treated by an osteopath at least once [1]. This is also why the Italian government decided to regulate osteopathy and include it in the authorised healthcare professions with the publication of Law 3/2018 [2], and the professional profile was recently approved by the State-Regions Conference [3].

In order to prevent alterations of the musculoskeletal system, the areas of activity and competence of the professional profile include: (1) osteopathic assessment through observation, perceptual palpation, and osteopathic tests to detect the presence of clinical signs of somatic dysfunction (SD) in the musculoskeletal system; (2) osteopathic treatment by selecting purely manual osteopathic approaches and techniques appropriate to the patient and clinical context; (3) osteopathic treatment outcome assessment, verifying their appropriateness, planning follow-up, and sharing them with the patient; and (4) patient education for proper self-management within a multidisciplinary perspective.

SD is defined as “impaired or altered function of related components of the body framework system” [4]. Although SD is much debated in the osteopathic community of practice regarding its reliability and validity [5,6,7], it is classified in the International Classification of Diseases as a biomechanical lesion not elsewhere classified, proposing different body regions as possible locations [8] and it still remains a defining element of the profession: models of structure and function relationships, to interpret the meaning of SD within the clinical context, guide osteopathic practice in diagnosis and treatment [9]. SD is detected manually by osteopaths through specific clinical signs and palpatory findings (PFs): tissue texture abnormalities, positional asymmetry, restricted range of motion, and tenderness, which are summarised by the acronym TART [10].

The definition of the professional profile induced the Italian osteopathy community to investigate how osteopaths work. Surveys, such as that conducted by Cerritelli et al. [11], have reported that the Italian osteopathic community shows differences within their clinical approach, especially in diagnostic modalities. The Italian Register of Osteopaths (ROI), the most representative osteopathic professional association in Italy, has recently produced an Italian core competence framework in osteopathy to highlight possible professional competences, e.g., the importance of formulating osteopathic diagnostic hypotheses according to osteopathic principles and models, identifying the clinical relevance of SD or other useful clinical outcomes in osteopathy [12]. Moreover, the Italian osteopathic community of practice has shown a diverse theoretical and practical understanding of osteopathic models to guide clinical reasoning [13].

New hypotheses have recently been formulated by Italian osteopaths which investigate the meaning of palpation in the context of evaluation and diagnosis in osteopathy. They recognise the importance of tissue palpation in assessing patients’ health, considering body regulation (homeostasis), adaptation to stress (allostatic load), and inflammation [7,14,15,16]. Hands-on interventions can evaluate a patient’s responsiveness as well as stimulate the integration of proprioceptive-interoceptive systems achieving better body awareness [17,18,19,20,21,22]. PFs are one of the multidimensional aspects that may inform osteopathic decision-making within a biopsychosocial (BPS) person-centred model [23,24,25,26].

Therefore, considering these new perspectives and the heterogeneity of Italian osteopaths about hands-on manual approaches for assessment and diagnosis, the aim of this study was to investigate how a selected group of Italian experts use PFs in their clinical practice.

## 2. Materials and Methods

A qualitative inductive approach with a constructivist paradigm using data analysis proposed by Corbin and Strauss in grounded theory (open; axial and selective coding) [27] was chosen to describe how Italian osteopaths use PFs in clinical settings. The recognition in Italy of osteopathy as a healthcare profession within higher education could lead to changes in professional identity. In this sense, the approach and paradigm used aim to describe one of the phenomena characterising osteopathy, namely, assessment and diagnosis, by identifying the influential factors starting from the informants’ experiences. For this reason, participants who were both clinical and research experts were selected to describe the phenomenon, taking into account knowledge and skills considering evidence-based practice (EBP). Due to the flourishing Italian scientific production of the last few years on the research question, a virtual focus group (VFG) via Zoom [28] was chosen as the data collection method [29,30]. The Italian promoters of this cultural growth were therefore able to take part in an interactive discussion, exploiting the possibilities provided by the conference call, typical of the historical moment.

The topic was not meant to be saturated in one discussion, and the aim was to create an enthusiastic group of peers willing to discuss. An inductive thematic model was used for saturation, focusing on data analysis and relating it to the new emerging codes or themes [31]. An initial anonymous survey (Appendix A) concerning the topic was filled out by the participants before the VFGs began, and a total of four VFGs moderated by the researchers were conducted throughout June 2020 to January 2021, once every two months. The survey results were used to initiate the fourth and final comparison.

This study was approved by the Institutional Review Board of SOMA Istituto Osteopatia Milano of Milan for informed consent from research participants on data protection and privacy as required by the European Union General Data Protection Regulation.

### 2.1. Recruitment of Participants

A purposive sample was selected among the most influential osteopaths with at least 10.000 h of experience within the osteopathic clinical, educational, and research fields in the Italian community of practice [32]. Seven of the participants were authors of papers concerning the topics explored in this research. Eight males and three females with ages ranging from 32 to 59 (43.9 ± 8.7) were recruited and approached by two of the researchers via email addresses available on the Italian Register of Osteopaths. All generations of Italian professional osteopaths were included, and attention was drawn to include a wide range of osteopathic education institutions (OEIs) of origin. All accepted to participate in the first VFG on Zoom, scheduled for June 2020. Everyone (12 participants; 2 moderators) was added to a Whatsapp group in which personal introductions were shared and the group started to bond. A certain amount of prior relationship between participants was inevitable given their position in the community of practice. Purposive sampling according to criterion was employed, the high number of experts recruited initially was intended to assure a minimum of eight participants at each VFG as qualitative research experts indicate as being appropriate [33,34].

### 2.2. Data Collection

The choice of conducting VFG allowed the participants from all over Italy to attend the meeting in a neutral environment with reduced and accessible costs.

Informed consensus on the use of data for research purposes and privacy policies were signed and collected prior to the study commencement.

The research team comprised one female (FB) and two male osteopaths (MG, FT), all being practising professionals and having different academic roles in Italian OEIs and interests in the beliefs and use of PFs in the osteopathic management of patients. FB had prior experience of qualitative interviewing, while MG and FT had conducted a project involving students in focus groups discussing different osteopathic thematics. FB attended certified courses in qualitative research in the medical field. Two researchers were present at each VFG discussion, with one facilitating the discussion; one managing the technical aspects involved, taking field notes, and summarising the topics. The third researcher was, therefore, recruited in order to study independently, and from a neutral perspective, the recorded discussions with the aim of also offering feedback on the interactions of the moderators with the participants and capturing non-verbal information. VFGs were audio and video recorded and lasted no longer than 2 h. The discussions were transcribed verbatim and returned to the totality of the participants after each encounter, even if not all had attended the discussion. The VFGs were conducted in Italian and the related discussions were transcribed into Italian and later translated by the authors into English in order to include them in the quotes identified and reported in the results. The participants were invited to check the accuracy of the transcripts. Having the researchers’ roles both in the educational and research fields, different relationships to a different extent with some of the participants were present prior to the study commencement. All participants received a recruitment email in which the goals of the study together with the professional area and role in the research of the two moderators were explained.

The four VGFs were constructed and planned differently depending on the data and interactions that emerged from the previous ones. Table 1 summarised the facilitating elements and questions used to create discussion. Each VFG was developed considering the latest evidence available; the data collected with the survey; the ongoing analysis of transcript data; and constant discussion and sharing of ideas between the three researchers.

The first VFG was designed of creating open debate and confrontation in a brainstorming atmosphere throughout the use of three photovoices with the aim to create initial disclosure and comfort in interaction with other participants; the second had its target on the use of PFs; the third was focused on the use of SD in the clinical setting; the fourth had the relation between PFs and clinical outcomes as the central topic of discussion.

### 2.3. Data Analysis

The initial survey gave the researchers an approximate idea of the range of opinions present in the group about PFs, clinical outcomes, and patient management in osteopathic clinical practice. Thus, ensuring a heterogeneity of arguments to discuss and giving the moderators an idea of how much could be ‘pulled out’ of the discussion (e.g., opinions on the usefulness of PFs in the survey ranged from ‘none’ to ‘fundamental’).

The unit of analysis was the entire content of the transcripts of the four VFGs obtained by adding the results of each meeting to the others. Potential identifiers were removed from the transcripts prior to data analysis. Participants were asked to provide feedback on transcripts but not on the findings.

A qualitative description of the VFGs transcripts was performed by all three researchers following the three-phase analysis and coding described by Corbin and Strauss [27], i.e., open coding, axial coding, and selective coding, respectively, to analyse the data in all possible ‘directions’, to search for the relations within the data, and to select the core category to relate it to other categories. Categories are the basic guide through which themes, the final step in the coding sequence, are created. Categories are formed by comparing and contrasting the concepts, which are interpretive words that group the codes that share similar ideas. The code is the first item that emerges, and it is the label given to the data extracts that have meaning.

The researchers used an audit trail and multiple coders to achieve analytical rigour; the coding team comprised an experienced clinician with 25 years of experience in clinical practice and was active in the process of professional recognition in Italy (AB). The team identified and validated emergent themes through an iterative process of listening and debate to mitigate the potentially deleterious effects of preconceptions.

The programme Quirkos was utilised to catalogue, share, compare, conduct memoing, analyse results, and generate codes from which themes were derived. Quirkos is a CAQDAS software package for qualitative analysis, designed to help sort and manage text-based data, by managing sections of text described as being about a particular topic or theme [36].

The themes were developed without external influences, such as theoretical perspective or framework. Moreover, there were no particular contrasts to be managed with respect to the observations produced by the participants and the themes identified.

## 3. Results

The participants (*n* = 12) of this study had an age range between 32 and 59 (M = 43.9), with the majority of the sample being male (75%).

The purposive sampling criterion identified participants distributed throughout the country. They came from similar osteopathic educational backgrounds, most had studied osteopathy after a degree in a healthcare profession (part-time or Type 2 programme) and some graduating directly into osteopathy (full-time or Type 1 programme) [9,37].

The average number of years in their osteopathic clinical practice was 12.7 (Table 2).

A total of four narratives describing the direct testimonies of Italian osteopaths interviewed regarding the use of PFs in the context of their clinical practice were collected.

Four themes were developed as a result of the analysis of the data gathered: (1) osteopathic identity; (2) evaluation; (3) osteopathic diagnosis; and (4) sharing. The findings in this study are illustrated in the coding tree in Figure 4.

Qualitative descriptive research about how a selected group of expert Italian osteopaths use PFs in the clinical management of patients shows that the professional identity of osteopaths is characterised by the use of the hand and, therefore, by the role of touch and manual skills, characteristics which are dependent on the subjectivity of the osteopath. The osteopath contacts the patient’s tissues by applying external forces with the hands, engaging in a relationship that leads to detecting PFs in the evaluation. The assessment of a complex system has to consider numerous variables that lead to an osteopathic diagnosis, such as clinically meaningful PFs and clinical reasoning. PFs relevant to clinical practice depend on shared decision-making and SD. Sharing the results obtained from manual evaluation with oneself, with colleagues, with other healthcare professionals, and with the patients, while maintaining its distinctiveness, should become shareable using understandable terminology and clinical outcomes.

Narrative descriptions of all the themes are presented in the following section. Quotes from the focus groups illustrate opinions throughout the findings section to ensure transparency and close connections between data and findings. All quotes used for qualitative analysis can be read as Appendix A. The most representative ones are given below.

### 3.1. Themes, Categories and Subcategories

#### 3.1.1. Osteopathic Identity

Professional identity is correlated with touch and manual skills, which are operator-dependent (Figure 4).

Before diving into the main topics of research concerning PFs, some common features emerged, drawing a baseline on which the rest of the discussion could be built. Differing viewpoints regarding touch and manual reliability suggest that the topic has yet to find full agreement throughout the profession, with different points of view emerging. The role of touch was connected to different themes, but for the means of this research, only the one that relates to PFs will be exposed.

##### Professional Identity

The recent professional profile approved by the Italian government defines osteopathic evaluation as depending on the so-called ‘perceptual palpation’. Throughout the focus groups, this theme was largely debated, and one fundamental element emerged: osteopathy is a manual therapy, and as such, it is mandatory for it to have a manual component in which touch, palpation, and the perceiving hand are of primary importance.

The following quotes highlight this concept:


*BG: “… I think that for osteopaths the hand is fundamental so the idea of abandoning something of the profession, I don’t know what it could be but certainly not the palpatory aspect, I think this is very important.”*



*VL: “… we can’t lose palpation. Actually, I think we have already lost so many things, I wouldn’t leave anything to other professions, we already gave away a lot.”*


This debate was created following what emerged from one of the informers who was questioning the importance of palpation in the osteopathic profession:


*PG: “… maybe it would be important to weigh, give a weight to this main theme of palpation […] we have to find a tool that allows us to characterise our profession, that is univocal. Surely the aspect of perception does characterise it but it is very unreliable and I would therefore put it on a second level, I wouldn’t throw it away but I don’t think it is of primary importance.”*


The discussion also highlighted a range of words and ways of saying that the informers used to describe the so-called perceptual palpation.

The following quotes show this issue:


*BD: “… a perceiving listening, which is the ability to listen to the tissue but it is actually a listening to the person in general …”*



*CF: “… the perceiving touch, potentially is something we can include within a sensation or a somatosensory analysis of what we perceive with touch.”*


Some informers firmly sustained that this is actually a big issue that should be addressed and that it reflects the uncertainty of the act. The topic of metaphoric language was addressed. This can be seen in the following quotes:


*CG: “… this osteopathic jargon for which we use “to listen to” that is evidently a synesthesia because one doesn’t listen with one’s hands, I listen with my ears […] when we tend to use a figure of speech to say something because we are not able to define it differently […] it empowers the fact that we are all confused.”*



*LC: “… actually it is an Englishism in the sense that some authors use a metaphoric language ‘to listen, listening, general listening or local listening’ […] that also means to consider, to pay attention […] we are talking about perception which is in the domain of touch, the communicative aspect in the touch.”*


##### Operator Subjectivity

As the role of the perceptual hand gained importance and was elected as a characterising element of the profession, discussion on what generates, creates, and fine tunes a “perceptual hand” opened the stage to the crucial aspect of subjectivity related to an operator-dependent act.

The following quotes highlight some of the elements that are considered important in terms of how the individual osteopath perceives them. The background elements that emerged are outlined below:


*BD: “… surely related to the educational curricula of the person but not necessarily those that are the technical competencies but what is really the background of that person, intended like the journey of that person at a 360° […] the journey of personal growth … as a human being.”*



*DFF: “… we can surely have a technical toolbox that we acquire during our educational path, we can train it with different schemes […] then there’s this big slice made up by the ‘Self/I’; the ‘I’ operator with its kind of experiences not only technical osteopathic, but of all kinds.”*


Other individual elements concerning everyday life emerged as being important in creating the perceiving experience:


*CF: “… depending on how the operator is interacting with the area from a cognitive point of view, meaning if he is concentrated, if he is paying attention or not paying attention, if there are elements that tend to influence him, if he is distracted or not distracted. How many patients he visited, the context in which he is, how he is feeling.”*



*BD: “… it considers all of what the patient told me before … all of what the patient talked about from the moment he/she entered the office …”*


An interesting focus was also put on the cultural background of the operator:


*CF: “… the culture of the operator …”*



*DFF: “… even cultural aspects […] there is a cultural aspect, the palpation of an Italian osteopath could differ, even just for this aspect, from the perception of a British, Indian, Australian, Native American.”*


A common endpoint was the fact that all these elements put together lead to a biased/subjective outcome of the palpatory assessment and its reliability.

These are some of the quotes:


*DFF: “… from the moment we enter the sphere of perception of a person, we open thousands of windows even the unthinkable that make the person unique therefore the elaboration that is driven by the touch becomes unique …”*



*CG: “… we put together a whole series of things for which we nearly have a mathematical certainty that a perception can be biased […] we know that our perception by itself is biased.”*


#### 3.1.2. Evaluation

The osteopath contacts the tissue of the patient by applying external forces and engaging in a relationship that leads to detect PFs (Figure 4).

One of the main topics of the research was to define how Italian expert osteopaths translate their personal beliefs, knowledge, and experience into their clinical practice. In these terms, the categories that emerged from the discussion regarded the role of the osteopath in patient healthcare, the main forces that they use during osteopathic clinical practice, and the importance of establishing a good therapeutic relationship.

##### Osteopath

As mentioned above, the osteopath’s subjective background plays a central role in osteopathic clinical practice. Osteopaths, with their experience, have the task of engaging in a multidimensional relationship with the patient and their tissues, with the aim of reaching an osteopathic diagnosis, the milestone of osteopathic care.

Regarding this issue, the following citations underline the participants’ thoughts:


*CF: “… if we look at it from the operator component it means that the perceptual information and therefore of the touch that arrives at the higher centres of the operator’s brain tend to take two paths and this is physiology […] The two then integrate within the operator’s brain which in turn creates perceptual maps that are the synthesis of what the operator is feeling at that moment or thinks he feels at that moment plus the operator’s history.”*



*DFF: ”We can say everything we want, that ‘self/I’ -operator- part has a fundamental weight, despite the technical aspects and surely there must also be the relationship with the patient.”*


Among the topics discussed regarding the operator, particular mention went to their culture, experience, and everyday life. These issues were described above in the operator subjectivity section. 

Force

Through external forces, the osteopath induces stimuli that investigate the patient’s tissue. Participants referred to this phenomenon in several quotes that are reported as follows:


*BG: “… about how we use palpation or the hand in daily practice, I can tell you what my experience is, which is basically based on the use of an extrinsic force on the tissue. A force that is dosed according to what it is the request you want to make to the tissue …”*



*BA: “… I use my hand to understand which forces to use.”*


The types of forces are divided by participants into two different types, forces with assessing scope and therapeutic forces:


*DFF: “‘Forces, forces at play’ right? An aspect that is part of our evaluation phase […] but then that guides our entire therapeutic practice …”*



*BA: “So listening is not a gesture, a passive reality, that is […] but I give information, a small, large input, etc., a force, the patient’s reaction is what guides me in the evaluation, in the information that is useful to then decide to do something.”*


The debate also covered the types of forces used in the evaluation phase to detect specific tissue characteristics and the capability of the operator to manage these forces and their response:


*BG: “… the use of an extrinsic force on the tissue, a force that is dosed according to what is then the, let’s say, the request you want to make to this tissue, i.e., if you want to simply evaluate an aspect of consistency, an aspect of temperature or if you want to evaluate an aspect of motility and if you want to go for a motility evaluation, the extrinsic force that I use is an isometric contraction of the deep flexor muscles of the hand.”*



*CF: “In general it depends on the type of touch that one wants to use, but basically what I use is actually a force so it is a pressure that the operator induces at different levels, we want to call them Newton, at different Newtons, different pressures with a different force on the various parts of the body.”*


##### Patient

In a healthcare process, it is important to consider all the ‘actors’ involved. From the participants emerged the importance not only of the operator and the forces imposed by them, but also of who receives these stimulations and who entrusts themselves to the treatment: the patient. The following quotes report the importance that osteopaths give patients.


*LC: “… the insertion of the patient in the process is a step that we must consider …”*



*CG: “… 50% is what we touch and the other 50% is the patient who is touched …”*


Participants agreed that the interface between the operator’s forces applied, and patient is represented by the body tissue:


*CG: “… the patient responds with tissue, it is a tissue response…”*



*CF: “… the contact with the patient that occurs through the positioning of the hand on the patient’s body area generates in turn a contact, therefore skin-to-skin, which in turn generates information for the patient himself and information for the operator…”*


Tissue

Once the role of the operator was ascertained, another element emerged as crucial during the evaluation process: tissue characteristics. Tissue is considered the somatic entity subject to palpation. Belonging to the musculoskeletal jargon, it refers to the connective tissue and its rheological characteristics, influenced by the vascular/lymphatic and neurological systems; a physiological state of the tissue coincides with a good function of all the systems.

The following citations highlight how some of the elements concerning the tissue perceived by the individual osteopath led to decision-making:


*CF: ”… identify one of the salient characteristics of the tissue that trigger a decision-making mechanism on what the operator is perceiving …”*



*CG: “The rationale for this type of approach is basically when “I like-dislike” an area, and it becomes interesting from a clinical point of view [..] from a palpatory point of view then the tissue responds to me so it has a reactivity capacity […] then my hand tells me this interests me now …”*


Concerning the main features which osteopaths look for during practice, participants report reactivity and movement of the tissue as being the main elements: 


*CF: “Personally I believe that the quality of the tissue and therefore the response of the tissue -if we want to remember the TART, if you like it-. The tissue texture alteration, I believe that it can be the key element that triggers this reactivity on the tissue. On the other hand, anything that I personally happen to feel-as I was taught- is the reactivity of the tissue as the main element …”*



*BA: “… movement as an important variable in the evaluation we make and I speak of a movement that has to do with very small ranges […] what I personally use cannot ignore, in my opinion, the movement.”*


These differences reported are not completely in contrast, indeed:


*BA: “… the concept of tissue consistency, which is always registered through pressure, so as CF said, we speak of Newton, of kg weight, that is, I squeeze something, or pull something … actually I, for me we are always talking about movement.”*



*TM: “… following a logical pathway but in a more macroscopic context […] I first evaluate the movement and then I entrust myself to the palpation; or at the same time through the palpation I also evaluate the movement.”*


However, differences between the tissues’ characteristics, and beliefs are emphasised. The following sentences underline these discrepancies:


*CG: “… responsivity of the patient which is partly what CF says because the patient responds with tissue, it is a tissue response, call it thixotropic, call it whatever you like …”*



*VL: “We see osteopathic diagnosis […] listening to BG and CF in a slightly different way; in the sense that for me palpation is, what I evaluate with palpation, it is not so much a question linked to pressure but to the movement. I feel it is important to analyse the movement, if we put it in simple terms, of the micro movements or -as you know, we have described it- a movement in the neutral zone. So for me, palpation is first of all analysing how the system moves.”*


##### Relationship

Analysing the roles of osteopath and patient, the participants also focused on the relationship between them, considering it fundamental in the decision-making process that leads to care. Below are some quotes, which present their opinion about that:


*LC: “… is the relationship with the patient or if you want the patient, maybe I prefer the term ‘relationship’, then in the shared decision-making process that is defined as the use of palpation.”*



*CG: ”… we no longer even talk about person-based medicine but relationship-centred medicine. Therefore, it is no longer an operator centric, it is no longer patient centric, it is a relationship centric. And this here is very interesting, it is very interesting because the patient’s responses, the patient’s sensations, I use in the treatment.*


In this kind of relationship, the communication channels were discussed by the participants. A particular mention concerns the type of dialogue mostly used, verbal or non-verbal: 


*BA: “If you like the touch, if you like it too much … how should I change my touch but the dialogue tends to be manual, non-verbal …. […] therefore the verbal dialogue yes but the ‘dialogue’ if I had to put it on the scales, it’s definitely more manual than verbal … […] I speak but with my hands.*



*DAG: “The relationship is present within the question that I imagined myself asking the patient, if I were there as an operator: “what do you feel?” So, in any case the patient is there quietly, both palpatorially and verbally.”*


The considerations regarding the patient/operator relationship enhance the attention regarding the centrality of the patient during osteopathic clinical practice:


*CG: “So, I don’t know, the thing that I would like to add in an important way is precisely this active and not passive presence of the patient during the whole process, which can be both the evaluation and the therapeutic one afterwards.”*



*DAG: “… what I have begun to include in my clinical practice in recent years is the patient’s perspective, which is the subjectivity that osteopathy then already has also in the TART criteria with T, with tenderness. But the patient’s perspective has become a little more important.”*


Palpatory Findings

Considering what was mentioned above, osteopaths engage in a relationship with the patient mainly throughout palpation. From this touch-mediated relationship emerge findings that are characteristic of the variables detectable by palpation, such as texture and movement of the tissues:


*BG: “… a more local aspect, we can define it as a palpatory observation of the characteristics and properties of a tissue in that case therefore taking into account the consistency, density, presence of resistance, barriers in this sense we speak of perceptive listening.”*



*VL: “… the impaired function […] I consider it, let’s say, an adaptation of the system to something that is required of it; and I evaluate this through movement.”*


#### 3.1.3. Osteopathic Diagnosis

The assessment of a complex system has to consider numerous variables that lead to osteopathic diagnosis (Figure 4).

The discussion further led to the concept of osteopathic diagnosis. An assessment of the patient, including PFs, is considered within a clinical reasoning process that should lead to a shared diagnosis.

The theme of having to deal with a complex system was addressed, as the following quotes show:


*LC: “… when we talk about a complex phenomenon, it is very difficult to find just one test […] there has to be a series of elements we put in. One day we will have to decide how as well as which (elements) we should use.”*



*CG: “… we are talking about the assessment, but since we are in a complex system, the relationships between elements can be observed only retrospectively […] I doubt we have the possibility to verify if our decisional process is coherent or not […] trying to find direct links between cause and effect is a waste of time.”*


Osteopathic diagnosis is expected to consider the patient as a whole and is therefore an ongoing process that evolves the following:


*BG: “…There is a specific diagnostic moment that then needs to be amplified, or completed, by a much greater assessment or relationship with the patient to make a real and proper Osteopathic Diagnosis.”*



*LC: “Another bias of mine is “considering osteopathy centred on people”, and if it is centred on the person, the decision-making process is shared. And in a shared decision-making process, if I touch a thing and decide how to treat it myself … non-verbally … that is, because that’s another gap, that is, I touch, and then I want verbal feedback … then we have to decide. It is probably my cognitive bias but it is written all over the place, which is centred on the person, and today we say “about the relationship”. So I start from this assumption …”*


The importance of involving the patient in the diagnostic phase emerged also as a hypothesis for limiting operator-dependent bias. The following quotes highlight this:


*LC: “ … inserting the patient in the process is something we need to consider …”*



*CG: “We know our perception alone is biased is the thing that makes us want to combine it with someone else’s perception of this thing.”*


To establish the importance of involving patients in osteopathic diagnosis, participants focused their attention on defining what they consider an osteopathic diagnosis:


*LC: “… in a shared decision making process to have process addressers which are based on a verbal feedback from the patient just considering the body awareness of the patient which can be labile, that is it can be more or less active, we need other process addressers which can be subjective as well, […] in my opinion linked to the disorder, linked to the perception, to the sickness, to the experience of the disorder or of something he or she considers related to the disorder and when we go to the comparable symptoms it is also far away.”*



*TM: “… I (osteopath) relate all the different dysfunctions that I have been able to bring to light, then I relate them to the reason for the consultation and try to make a synthesis that is steeped in clinical reasoning and is based on my knowledge, experience and then I decide which and what to treat…”*


##### Clinically Meaningful Palpatory Findings

Regarding this topic, as just mentioned above, Italian experts wondered about the clinical relevance of palpation and SD. Not all of the participants agreed that SD is always relevant in their clinical practice:


*BG: “… that the somatic dysfunction has its clinical component, its semeiological component, is out of the question in my opinion.”*



*LC: “… who may or may not have it (referring to clinical role of SD) … in fact we speak of “severity” of the dysfunction, of “clinical relevance” of the dysfunction.”*


Shared Decision-Making

According to some informants, PFs correspond to ‘neurologically active’ body areas, which are used by osteopaths to convey the effects of touch and active strategies of person-centred osteopathic care.

This approach allows sharing clinical reasoning and treatment decision-making with the patient, which are fundamental in a person-centred therapy that aims to stimulate body awareness for healing purposes:


*LC: “… in the shared decision-making process the patient has a part because he or she checks if what I am touching has an impact on his body functions, which he or she recognises with his or her body awareness … the ‘familiar symptoms’, the patient is aware of them, he or she tells you; the ‘signs of comparison’ are those where the osteopath accesses through an intuition and a palpation to some aspects that are not accessible to the body awareness of the patient in that moment and therefore the verbal feedback arrives through the mediation of the palpation of the osteopath.”*



*DAG: “… Then about the role of the patient … for example we can make the patient aware of an area that he or she may not consider …”*


Somatic Dysfunction

PFs prompted debate about one of the entities characterising osteopathic diagnosis: SD, with regard to its definition and clinical signs.

Somatic dysfunction is described as an alteration of function and not an alteration of structure:


*LC: “… somatic dysfunction is a compromised function and an altered function, and in osteopathy we also describe what these functions are and that it is related […] to a body framework, therefore to the soma … a dysfunction is not a dys-structure, it is a related alteration of function emerging in a body region or pattern. … It’s not an altered soma related to an altered function, it’s an altered function …”*



*BG: “… when making a diagnosis we generally have to consider a clinical aspect, a semeiological aspect, an aspect of pathophysiology […] The moment we are talking about altered function, we are talking about a clinical aspect.”*


According to the informants, this alteration concerns the ‘somatic function of the whole’ of the body framework system, which provides an environment that allows all the systems of the body to function in an integrated way:


*LC: “… it seems that this entity that we (osteopaths) palpate represents something in the soma to convey the effects … related to the relationship between operator, patient and environment.”*



*DAG: “Somatic dysfunction, which was already for me a gateway to the patient system.”*


According to informants, the presence of SD can be related to the health and adaptive capacities of the patient:


*VL: “… when I palpate the dysfunctional part […] for me it may not be related to the problem but it may be very much related to the adaptation that the system has for a problem or a pathology …”*



*BA: “… the altered function of body systems, where, in my opinion, it can be placed in the context of adaptation, in the health of the patient as an adaptive capacity …”*


They agree that clinical signs for SD, identified to date, are not reliable and scientifically proven to be valid:


*BG: “… we can argue that we don’t yet have the tools to measure it, that we don’t yet have interoperator reliability …”*



*DFF: “… not being sure yet to detect something reliably and in a clinically meaningful way, then it creates space for further interpretations, enlargements and whatnot … there is the problem of reliability, there is also the problem of validity and so I say to myself: beyond, without wanting to trample on our perceptive self of which we have spoken before in abundance, but is there a common trace, a method, an operative way, which is more reliable than what we have now and which we have seen to be unreliable? Is there a diagnostic entity that we are all calling somatic dysfunction, that is a little bit more clinically relevant, a little bit more clinically valid than the one we are adopting now?”*


Regarding how to identify possible SDs relevant to treatment, the informants, however, expressed two different points of view: (1) the importance of tissue quality and (2) the value of movement quality.

For some, the quality of the tissue palpated, expressed by its texture and tenderness/sensitiveness, is evaluated to gain information about the responsiveness of the patient’s tissues:


*TM: “… to the concept of somatic dysfunction certainly an initial clue is given to me by the tissue or tissue changes and then these tissue changes attract my attention …”*



*PG: “Personally I give a lot of importance to the provocation of the symptom, in addition to what has already been said, so I try in some way when a patient comes to stimulate the area that could be the source of the pain, so the manifestation of the symptoms for me assumes an important meaning because then it directs me to the region and the area to be treated.”*


For others, a characteristic clinical sign to identify SD is movement and its restriction. In particular, not the amount of movement, but the quality of movement of a certain body portion, which, as per definition, in a dysfunction presents a more restricted and a freer motion:


*BA: “… this motor variability is for me the clinical sign that I use in my clinical practice, … I am always looking for that aspect, that is the free movement or the restricted moment, this variability is what I look for in my practice to look for what I consider altered in the patient and I need it to find, correcting it, … “*



*VL: “… we see it a little bit differently, in the sense that for me palpation is, what I evaluate with palpation, is not so much a question of pressure but of movement. I palpate to analyse the movement, if we want to say it in simple terms, micro movements or as you know we have described a movement in the neutral zone. So, for me palpation is first of all to analyse how the system is moving, …”*


##### Clinical Reasoning

It is the logical process that the osteopath uses to understand the symptoms and signs reported by the patient. This process is the result of the evaluation, and permits the attainment of the osteopathic diagnosis:


*LC: ”… it is the diagnostic process and the therapeutic process, which is based on a shared decision-making process with the patient, that tell me how to use palpation, in terms of type of touch, approach or technique …”*



*TM: “Subsequently, after I have finished, I have exhausted all my osteopathic evaluation that makes use of palpation, I put in relation all the different dysfunctions that I have managed to bring out, subsequently, let’s say, puts them in relation with the reason for consultation and testing to make a synthesis that is imbued with clinical reasoning and is based on my knowledge, experience and then I decide if necessary what and what to treat […] palpation, possibly if I find some tissue alterations I test the movement and then memorise and then subsequently compare I evaluate what I find, I reason and I make a synthesis to understand what the most suitable therapeutic approach may be”.*


#### 3.1.4. Sharing

Sharing the results obtained from manual evaluation (Figure 4).

All participants agree to identify in sharing one of the essential processes related to the use of PFs in clinical practice and specifically in a multidisciplinary vision of patient care.

##### Recipient

The recipients of the sharing process have been identified by the participants as four entities: oneself, other osteopaths, healthcare professionals, and patients. Participants also highlight the importance of finding a way to share with others the internal dialogue that each osteopath has with themselves during their clinical practice:


*DAG: “… the outcomes are all important because if I relate to myself some are significant, if I relate to the patient others are important and if I relate to the doctor others still.”*



*LC: “… absolutely subjective internal dialogue and then I add a procedure that allows to explain to the world that subjective internal dialogue that has become a dialogue of the operator-patient relationship.”*


Osteopathic Distinctiveness

Participants stressed the importance of sharing information about the specificity of the osteopathic profession, which is currently not always understandable to others:


*VL: “The information I share with the clinician comes from a specificity of my profession …”*



*BA: “What we value has a specific meaning for ourselves and that is difficult to communicate to other professionals. They don’t understand it.”*


The following subcategories highlight two factors that are considered important for efficient sharing.

Terminology

Words that osteopaths use to describe specific processes and findings that they evaluate. All participants claim that osteopathic terminology is not always comprehensible to other healthcare professionals:


*BA: “… we (osteopaths) find it difficult to communicate with others because we use a terminology that others do not find in practice, in their studies and research […]. Anterior iliac codifies a pelvic somatic dysfunction. For us it is a code that can justify an aspect but in the context of other professions they don’t understand it.”*



*CG: “The problem is terminology deficiency regarding many of the things that we (osteopaths) would like to be able to describe on the musculoskeletal system.”*


Outcome

Participants identified the importance of using scientific indicators as an interface to communicate with other professionals:


*BL: “all of us have to face the international scientific world, based on scientifically measurable outcomes.”*



*DFF: “in my clinical practice I try to find a way to communicate with a clinician according to certain indicators”*


## 4. Discussion

The aim of this study was to investigate how a selected group of expert Italian osteopaths use PFs in the clinical management of patients, particularly in relation to osteopathic assessments and objective examinations.

The thematics emerged from the debate concerning osteopathic identity, evaluation, osteopathic diagnosis, and sharing with different recipients, with a large accordance among participants on the peculiarity and distinctiveness of osteopathic palpation in clinical practice.

Although Italy is regulating the osteopathic profession, aiming to insert it within the national healthcare system, professionals working in the territory show different attitudes and beliefs towards the actual practice itself [11]. Osteopathy is a very young healthcare profession in Italy, where the first generation of professional osteopaths is still in practice, teaching, and at the head of OEIs [37]. Tradition, with an emphasis on the role of the osteopath and their “listening, seeing hands” [38] is still one of the main features that characterise the way many professionals feel about themselves [39]. In confirmation of this, although Italian osteopaths are in favour of EBP, they lack basic skills in EBP and rarely engage in EBP activities [40], thus, maintaining above all a hands-on operator-dependent clinical approach [41], which is supported by some evidence [20,21,22], but at the same time has shown scientific fragilities in terms of reliability and validity considering the complexity of the phenomenon involved [7,42,43,44,45,46,47]. Osteopathic reliability can be related to perception and interpretation of the palpation which is influenced by previous experiences, type of information to collect, habitual and context-related influences, and cultural and social imprinting [42]. The same considerations are found in related professions, such as physiotherapy [48,49].

In Anglosphere countries, where osteopathy was born and has been recognised for decades, the manual component of clinical practice has been reconsidered over the years, with hands-off and BPS approaches gradually gaining importance and significance [50,51,52]. In countries where osteopathy is not recognised, there is a strong growth in osteopathic traditional medicine with a hands-on imprint [53,54]. This phenomenon is associated with the development and attention to EBP, generally more advanced in countries where osteopathy is fully recognised [55,56,57]. However, the boundaries imposed by university education may diminish the transmission of osteopathic tradition and principles in the skills of professionals [58]. This concern has, for example, led to a review of undergraduate education in the United States of America in order to recover professional characterisation [59,60].

Italy seems to have a position of its own, with the profession being in that territory only since the 1980s, when Italy was importing current practices first from France and then from England [61]. Nevertheless, Italy has one of the highest numbers of educational institutes and professionals in all of Europe [62,63]. The country to date counts approximately 14.000 professionals [64], with the continuous opening of private educational institutes providing heterogeneous training and graduating professionals with different educations and no mandatory continuing professional development (CPD) required, given the lack of regulation. On the other hand, hundreds of students graduating every year have quickly led to a turnover in the generations, showing a great interest in research and attention towards the international community with the will to refresh and renew professional concepts and theories in order for them to be evidence-based. This has brought Italy to be among the countries that publish the most in the scientific osteopathic field [65].

This is crucial in this historic moment in which evidence is needed in order for governmental regulation to proceed, but as the osteopathic profession is catching up with the integration of BPS aspects, patient-centred practice, and hands-off approaches [50,51,52], research is showing us another path where the centre is represented by relationship, and this relationship can be developed through touch, thus, bringing us back to the importance of the hand and the dialogue mediated by tissues [17,26,66,67]. Additionally, in physiotherapy, where the orientation to musculoskeletal pain is currently directed more towards a hands-off management of the patient, the hands-on approach is being re-evaluated [68,69,70].

In osteopathy, as well as in physiotherapy, touch seems to offer practitioners a means of communication, communicating care ‘beyond words’ and creating a space between practitioner and patient that expresses safety and security [71]; touching a patient in the area of a complaint can contextually build trust and affirm that the clinician has heard the patient [72]. However, considering also the emotional component that touching brings, this relationship can be potentially dangerous; additionally, for this reason, only trained and registered healthcare professionals should use this approach in the clinical setting [69,71]. Nevertheless, it must be maintained in order to not dehumanise the patient experience, which is increasingly characterised by an over-dependence of staff on electronic equipment and technology [73,74].

The touch-mediated relationship guides the osteopath to detect PFs which could be related to the patient’s allostatic overload [15] and their ability to self-regulate and adapt [7], essential elements for osteopathic diagnosis and therefore OMT.

In this context, the data that emerged from this study highlighted how PFs, for some, have a different importance in relation to clinical meaningfulness expressed also through clinical reasoning shared with the patient [25,26], while for others, the palpatory assessment, in regards to the quality of movement as a clinical sign of SD, plays a fundamental role, assuming a useful diagnostic significance for the choice of OMT [7,75].

This raises questions about the different roles given to the patient in the two points of view: one is active, given by multidimensional and multimodal approaches with, thus, a greater self-awareness of their health condition [76]; the other is more passive. However, recent evidence shows the active role of touch [20,21,22].

It should be noted that, to date, the osteopathic core curriculum lacks education in soft skills and hands-off approaches, not allowing the development of the management of psychosocial and contextual factors [48,76,77].

In this landscape that seems to show uncertainty, the results of this study cover these points by highlighting an inclusive paradigm in which both parts, purely biological and hands-on and psychosocial hands-off oriented, are used in clinical practice generating links between biomechanical and biopsychosocial approaches rather than polarised positions of difference or conflict [43,78].

Although non-verbal communication is a feature of manual and manipulative therapies [26,69,79], effective verbal communication is essential to establishing a good practitioner-patient relationship and to building the necessary trust with the patient to enhance the healing process [80,81]. There must be effective communication between patients and healthcare professionals, to attend not only to the disease but to also consider the patient’s experience with symptoms, the effect of the illness, and what matters most to the patient when establishing patient-centred treatment. Therefore, the ability of a healthcare professional to communicate effectively is essential for developing positive patient relationships, fostering a welcoming environment, and enabling patients to openly voice their problems [82]. Effective language and communication are also considered as important parts of osteopathic clinical practice. In the UK, Australia, and New Zealand, current osteopathic practice standards emphasise the requirement for osteopaths to communicate effectively to provide safe and effective care [83,84,85]. Thomson and Collyer [86], in a qualitative study on the interpretation of the language used by osteopathic students when treating low-back pain patients, showed that patients’ comprehension of their pain and participation in their own care were greatly aided by the type and nature of the language used by practitioners. The authors point out that emphasising the use of pathoanatomical terms and biomechanical metaphors may create negative thoughts and disengaging behaviours in relation to pain and care. Of the same opinion is Fryer [52], who indicates that the use of inappropriate jargon “may confirm the impression of a serious structural disorder in the mind of a fearful person, leading to catastrophizing, fear avoidance behaviour, and unnecessary dependency on treatment”.

However, the combination of both forms of communication present in osteopathy, the non-verbal one mediated by touch and the verbal one, can play a reinforcing role in practitioner-patient communication and significantly enhance the therapeutic relationship [87]. The bi-directional nature of this interaction has also been reported in both the osteopathic literature as well as the wider physical therapy research, where touch in combination with verbal direction is used extensively to communicate both physical and affective cues [88].

Collaborative practice in a clinical setting is often challenging due to differences in paradigms and professional languages, especially between the healthcare professions and complementary and alternative medicine, as well as osteopathy. Morin et al. [89] pointed out in a study on interprofessional collaboration between physicians and osteopaths, how the greatest difficulties in co-operation arise from the complications of translating the results of osteopathic palpatory assessment into biomedical terms and the characteristics of manipulative treatment using a common and scientific language adapted to current biological plausibility principles; furthermore, the lack of face-to-face interaction and formal and informal communication between osteopaths and physicians can lead to a lack of understanding regarding the scope of practice of each practitioner. In this sense, the experts participating in the focus group suggest using clinical outcomes as useful end-point indicators for interprofessional dialogue.

Although osteopaths should make the effort to adapt their language to the biomedical paradigm for better collaboration, the terminology needs to maintain the osteopathic professional characterisation. The specific functional assessment, the clinical signs to identify the altered function and the manipulative techniques adopted to facilitate the mechanisms of self-regulation and adaptation can be expressed with a more understandable terminology to improve interprofessional communication and to better standardised to facilitate research. In this regard, some focus group participants suggest using movement and its variability as a clinical sign to identify the altered function that osteopathy seeks in its assessment [7]. Furthermore, the lack of standardisation in osteopathy regarding assessment [75], OMT [90], sham treatment as placebo [91], osteopathic clinical reasoning [45], and adverse events [92], does not facilitate the significance level of osteopathic research findings, and thus dialogue between osteopaths and other professionals is needed. It should be noted that a shared osteopathic terminology [4], standardised osteopathic medical records for correct data collection [93,94], checklists, and a guide to report the characteristics of the intervention used in the best possible way are already available. One example is the template for intervention description and replication (TIDieR) [95], which is useful in therapies with a high degree of personalisation and variability, such as manual and manipulative therapies [96].

Focus group participants also emphasised the value of finding a way to communicate with others, the ‘inner dialogue’ that every osteopath has with themselves during clinical practice. Probably even this dialogue, fundamental for clinical reasoning, must be comprised of perception, intuition, and uncertainty [97,98,99], but also supported by concrete factors useful for making therapeutic decisions, e.g., by using palpation or joint motion testing, practitioners can determine the appropriate manipulative forces for treating [100], identifying preferential direction, and optimal strain patterns in osteopathic functional analysis as suggested by Standley in his in vitro studies of human fibroblast cultures [101].

The results of the study reflect person-centred care according to the BPS model adapted to osteopathy (Figure 5), in which the biological factors are related to a hands-on approach and the psychosocial ones are associated with a hands-off approach.

The osteopathic care, on the one side, includes touch-mediated assessment and treatment for a tissue-mediated relationship with the patient and non-verbal feedback; in this context PFs, are considered as diagnostic elements with which the osteopath interacts through a bottom-up approach. On the other hand, the hands-off osteopathic approach involves patient management procedures through effective verbal communication for agreement with the patient, such as therapeutic education, health-related advice, and self-management strategies for health promotion and prevention. In this case, PFs are used as indicators or moderators of the clinical process in decision-making shared with the patient, giving more importance to the top-down approach. However, as Bohlen suggests [78], both hands-on and hands-off approaches end up involving top-down and bottom-up dynamics. The complexity of clinical practice often considers these two sides as integrated by EBP, which comprehends the osteopath’s experience (knowledge, judgement, and critical reasoning), the patient preferences (personal and cultural circumstances, values, and priorities and expectations), and the best evidence available (external and internal). EBP and the best available evidence play a fulcrum role in the balance between the osteopath’s distinctive manual skills and the osteopath’s skills for effective communication with the patient, allowing for person-centred care. Education cannot fail to take this balance into account, where tradition, professional distinctiveness, and a biopsychosocial strategy for approaching the person with an informed EBP will allow professional development in a healthcare context.

One of the strengths of this study was represented by the participants selected who fully reflected the construct described above, where tradition and research are integrated. In addition, participants clearly brought out contrasting opinions, allowing for constant critical reflection and leading to an open discussion in which the participants themselves questioned their colleagues in order to better understand the different points of view.

One factor that may have influenced the limitations of this study is the professional relationships between the participants, which may have hindered the emergence of different points of view, divergences, and contrasts. Another limitation was certainly the difficulty and the amplitude of the themes debated in an online discussion, which affects the participants’ ability to interrelate and observe each other. This specific limitation has been partially overcome through the possibility of transcription checking.

Given that a targeted sample of osteopaths experienced in clinical practice, education, and research in the Italian community of practice was selected means that the transferability of the findings to the wider osteopathic profession needs to be established through further research.

## 5. Conclusions

This qualitative research shows that expert Italian osteopaths use PFs in clinical practice with a mixed hands-on and hands-off approach. Osteopathic distinctive manual assessment is maintained by integrating it in the context of person-centred care and making use of the best available information gathered from the scientific literature. The profession placed in the healthcare setting will have to fit into a multidisciplinary context by sharing its peculiarities with other professions using an understandable language.

Furthermore, it appears that professional identity is facing a transitional phase in which one looks to the future not yet sure what to leave behind in one’s past. In this landscape that seems to show uncertainty, the profession has a great opportunity as tradition and evidence coexist. The ability to maintain tradition in a context of informed EBP could represent the innovation of osteopathic professional identity.

In order to improve consistency, plausibility, generalizability, relevance, and expected applicability of PFs in clinical practice, osteopathic practitioners, researchers, and educators could participate in an International Consensus Conference using the results of this study.

## Figures and Tables

**Figure 1 healthcare-10-01647-f001:**
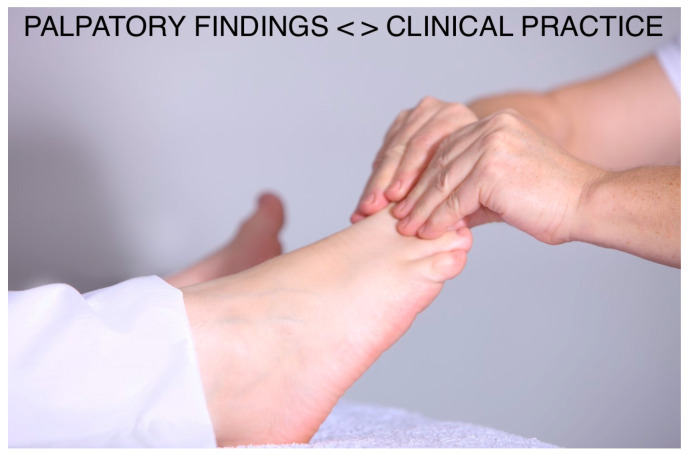
First photovoice: relationship between ‘palpatory findings’ and ‘clinical practice’.

**Figure 2 healthcare-10-01647-f002:**
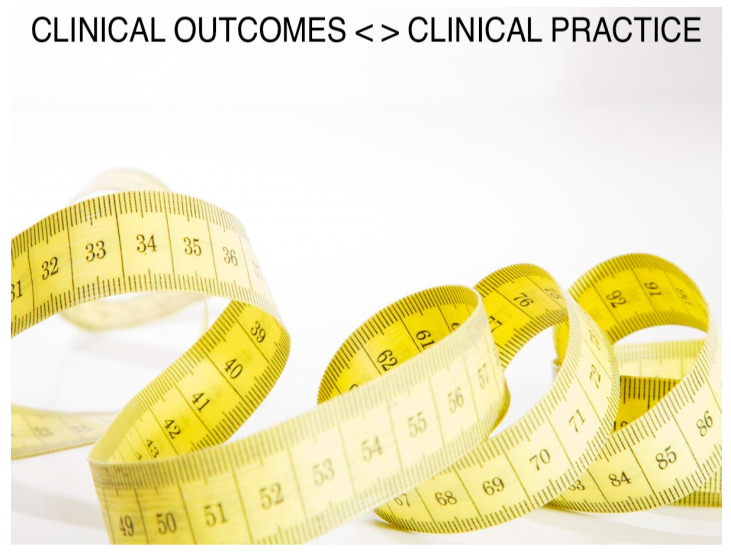
Second photovoice: relationship between ‘clinical outcomes’ and ‘clinical practice’.

**Figure 3 healthcare-10-01647-f003:**
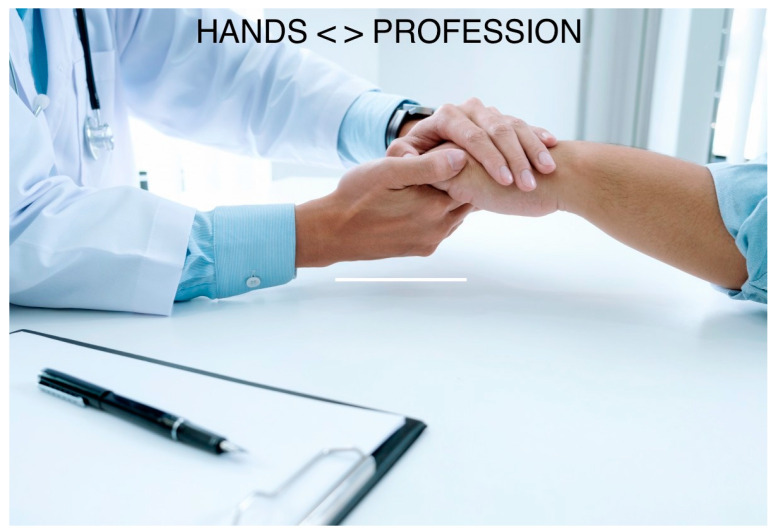
Third photovoice: relationship between ‘hands’ and ‘profession’.

**Figure 4 healthcare-10-01647-f004:**
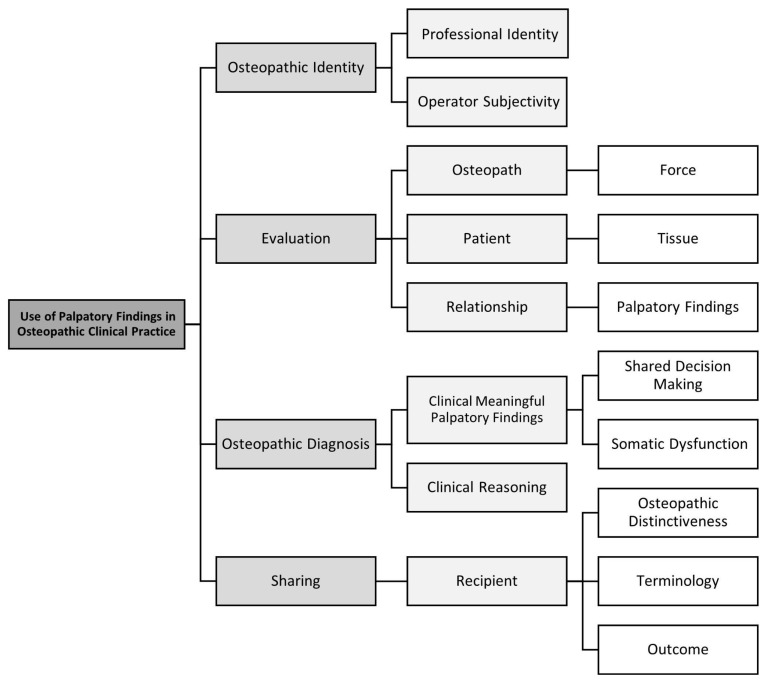
Use of palpatory findings in osteopathic clinical practice.

**Figure 5 healthcare-10-01647-f005:**
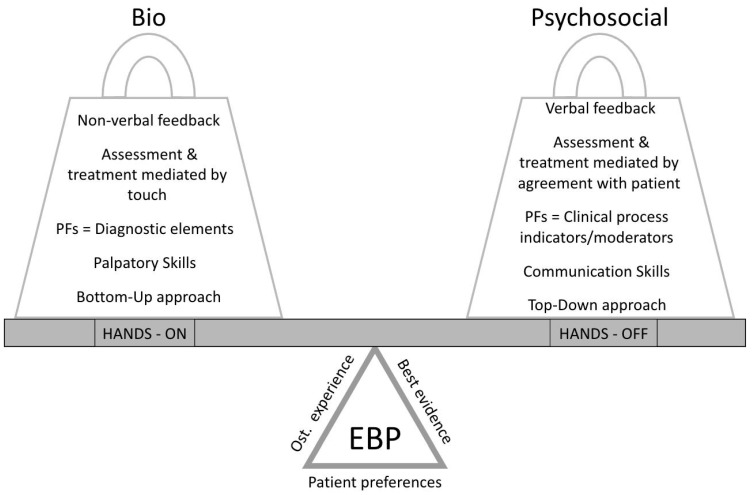
Evidence-based practice (EBP) plays a pivotal role in the balance of the osteopath’s competencies. PFs = palpatory findings.

**Table 1 healthcare-10-01647-t001:** Topic guide for virtual focus groups: facilitating elements and questions used to create discussion.

I VFG
The following photovoices were shown and discussed one at a time (Figure 1, Figure 2 and Figure 3). Participants were invited to observe the photovoice for 2 min and write down the three main topics that emerged in their minds related to the image. All the topics were then discussed and moderated by the researchers with prompts.
II VFG
Researchers investigated in depth and the recurrent term ‘perceptual hand’ emerged during the I VFG used to describe the way osteopaths feel and identify PFs. Two main questions were asked, and the discussion was moderated by the researchers with prompts: How important is the role of the ‘perceptual hand’ in your clinical practice? How do you use your ‘perceptual hand’ during your clinical practice, with your patients?
III VFG
The use of SD in clinical practice was thoroughly examined by researchers since this element emerged with contrasting opinions during the II VFG. The definition of SD [4] was presented and followed by a question: Do you use SD in your clinical practice? The discussion was moderated by researchers with prompts.
IV VFG
Researchers agreed that data saturation on PFs occurred, therefore, the last meeting concerned ‘clinical outcomes’. Participants received a file with useful information prior to and in preparation for the discussion. The file contained the definition and classification of the outcomes [35]. During the VFG, participants were invited to observe two charts that originated from the initial anonymous survey. The answers to the following questions were then discussed and moderated by researchers with prompts: Which clinical outcomes do you use in your clinical practice and how do you use them?

**Table 2 healthcare-10-01647-t002:** Demographic characteristics of the participants.

Age, mean (standard deviation) [min-max]	43.9 (±8.7) [32–59]
Gender, (%)	Male *n* = 9 (75%)
	Female *n* = 3 (25%)
Osteopathic educational backgrounds	Type 1 programme *n* = 3
	Type 2 programme *n* = 9
Years as osteopath, mean (standard deviation) [min-max]	12.7 (±6.9) [6–29]

## Data Availability

All data are fully available without restriction.

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
