# Peer review of "Beliefs and Use of Palpatory Findings in Osteopathic Clinical Practice: A Qualitative Descriptive Study among Italian Osteopaths"

_healthcare, 2022, doi:10.3390/healthcare10091647_

Round 1

Reviewer 1 Report

The authors have extensively studied the term “Palpation” in osteopathy. They have considered experts’ opinions in this field, operators, and patients-  also discussed this in the views of clinical practice. Their conclusion of advocating scientific indicators as an interface to communicate with other professionals evolved with a thorough study. One quick comment is that I recommend authors check the manuscript for spell and grammatical errors.

See the attachment for more comments.

All the best!

Author Response

Response to Reviewer 1 Comments

Manuscript number healthcare-1844832, entitled "Beliefs and use of palpatory findings in osteopathic clinical practice: a qualitative descriptive study among Italian osteopaths."

We greatly appreciate your readiness to have read our paper and to provide us with relevant feedback and useful suggestions to further improve the quality of our paper. A detailed description of all changes has been provided below.

One quick comment is that I recommend authors check the manuscript for spell and grammatical errors.

Response: Thank you for your comments. The text was checked for errors.

1) Many sentences are too long to read. It loses the essence halfway. Example: 18-21. I suggest authors go through the manuscript to break sentences for better readability.

Response: Thank you for your comments. Some sentences were shortened. The indicated sentence “There was a clear accordance among participants on the peculiarity and distinctiveness of osteopathic palpation with some divergency on the clinical meaning of the findings yet highlighting a complex multidimensional approach to diagnosis and treatment.” was modified as follows (lines 18-20): “Participants agreed on the peculiarity and distinctiveness of osteopathic palpation, but there was some disagreement on the clinical significance of the findings, highlighting a complex multidimensional approach to diagnosis and treatment.”.

2) Line 14: Change the word “territory” to “profession” or something more suitable. Line 275: his or her.

Response: Thank you for your comments. Line 14: we changed “territory” with “in Italy”. Lines 383-384: we have revised the sentence as follows “As mentioned above, the osteopath's subjective background plays a central role in osteopathic clinical practice.”

3) A brief description of what osteopathy is would be helpful to the readers.

Response: Thank you for your comments.  We have inserted the following sentence at the beginning of the Introduction part: “Osteopathy is a ‘person-centered’ approach to the prevention, diagnosis and treatment of illness and injury. Osteopaths are primary-contact health-care professionals that use a range of techniques including ‘hands-on’ manual techniques for assessment and diagnosis to treat various health conditions, as well as musculoskeletal structural problems that influence the body’s physiology.”

4) Table 2: Please use plus or minus between mean and (SD) in the last column. SD denotes somatic dysfunction as well. It would be more precise.

Response: Thank you for your comments.  We inserted plus/minus for standard deviation in the Table 2 and the abbreviation for the term “standard deviation” was removed to avoid confusion with the term “somatic dysfunction”.

5) Figures following figure 4 are not necessary. It is a subpart of Figure 4. Maybe the authors can find a way to recall figure 4- This would reduce the length of the manuscript (which brings me to my following comment).

Response: Thank you for your comments.  We removed the figures 5, 6, 7, 8 and only using figure 4 as a reference.

6) The manuscript’s length is too long – I suggest the authors use the quotes as supplementary information. As a reader, I am more interested in the outcome of the discussion – and quotes are a little distractive. Use one of two as an example in the main manuscript. I find the discussion is also circumventing and long. If the authors could shorten it to deliver the main points, it would make the manuscript more readable.

Response: Thank you for your comments.  We left only two meaningful quotes as examples and added an additional file (Quotes S1) to include those most useful for the analysis done. The discussion was shortened. After revision the pages became 22 from the 28 they were.

7) Also, Quotes, without the entire conversation, may not paint the complete picture. I would also like to see the results as a table or survey points (“none” to “fundamental”).

Response: The 'complete picture' is represented by the coding tree (Figure 4), which constitutes the analysis that emerged using the coding process proposed by Corbin & Strauss and summarily described in the Materials and Methods part of the article. The quotes are examples that argue the coding process that determines the themes, categories and codes, which are fundamental elements for the qualitative analysis of the text. All quotes are preceded by a brief summary of the thinking emerging from the panel of experts, where disagreements were present was reported.

8) I am curious to know if the operators faced any similar cases and discussed how different operates perceived the issue differently. Kind of like a controlled study?

Response: this study would present the results of a first study on the topic in Italy. The focus group represented the first situation in which the experts involved discussed the topic proposed by the study.

Reviewer 2 Report

Overview

The authors provide a thorough qualitative study that involves extensive input from osteopaths regarding several important themes in this profession that will inform future practice and research of osteopathy in Italy and elsewhere. I was interested to read it because I was involved in a similar project regarding the chiropractic profession, which is facing many of the same changes/issues. Therefore, I can see the parallels and how this material has value. I have a couple major comments, mostly to shorten the manuscript. I only recommended adding a few items. I had several remarks to improve the clarity of the manuscript. 

Major comments

1.       Length – I found the manuscript long at almost 30 pages. The introduction seemed to be a good length, as did the Methods. However, much of the excessive length seems to come from the quotes or excerpts. These might be abbreviated further. Also, in some sections you provided 3 quotes, in others you provided four or five. Maybe only 2 or 3 would be needed for each concept. Another idea would be to provide 2 or 3 quotes/examples then include the remainder as a supplemental file. However you approach this, I think you could trim these down where there is redundancy. Also, the Discussion is well written and contains a lot of good information but could be a bit condensed as it is over 5 pages. The conclusions should also be edited to a single brief paragraph.

2.       Results – In some of the results you provide a brief synopsis of whether there was either agreement or disagreement of the key concept or discussion/question/topic. In other sections, you leave it to the reader to sort through the quotes and determine for themselves what the main finding was. For example, in 3.2.1.3.1. Palpatory Findings, you state “The following quotes present the major opinions emerged…” I would prefer if you made sure each section provided a very brief summary, even 1 sentence, describing the key finding, or even if there is no clear consensus, that would be good to know. I think readers should be able to see the results quickly without needing to rely on reading the quotes.

Minor comments

1.       I think the introduction needs 1 sentence providing background on Osteopathy for those not familiar with it. For example, what type of health care provider an osteopath is or just a basic description (e.g., portal of entry, specialist, primary care, allied health practitioner), and what type of conditions they treat

2.       Maybe somewhere in the methods you should state if the interviews / focus groups were conducted in Italian then translated by the authors (e.g., your initials) into English? (if this is true).

3.       The supplemental file was a little confusing. The figures did not have any axis labels (# of participants, for example?). I’m not sure why the X-axis had 0.5 intervals if these were the number of respondents. Could you have 2.5 or 3.5? It seems all the numbers were whole numbers. Further, the bars could be sorted in order from greatest to least. Otherwise, it is difficult to interpret the figures.

Comments for clarity

1.       Abstract: “The Italian government has started the regulatory process for osteopathy to include it among the healthcare professions mentioning terms such as “perceptual palpation” and “somatic dysfunction” within the professional profile” – I was not clear what this means. Does this mean that that government will only regulate osteopathy as a profession provided that these terms are included within the definition of the profession? It is not clear what the government is trying to do to the profession with regards to this terminology. Based on reading this, I’m not sure why the government would care what terms are used by osteopaths. Also, I did not see this type of statement in the Introduction, aside from information stating that the profession is being regulated, so it was difficult to get more clarity on this.

2.       Abstract: “The thematics emerged were” – change to “Themes that emerged were”

3.       Abstract: “on the territory” – change to “in this field” or “in this profession”

4.       Abstract: “The results seem to reflect the history of the profession in Italy, which has evolved very quickly embracing tradition and scientific evidence” – I was a little confused by this statement. It seems that some participants embraced concepts with little supporting evidence, having more of a traditional viewpoint, in contrast to others which favored more recent evidence. Tradition and scientific evidence can be opposite things at times, so it seems too broad to say the profession is embracing both things. If the profession is evolving, to me that would imply traditional concepts may be left behind or abandoned in place of newer theories or research. Instead, it seems a little more complicated than that.

5.       Abstract: “The authors suggest further investigation to verify the state of art among professionals not involved in research or a broader consensus of the results to be proven.” – Maybe change “professionals” to “osteopaths”, and delete “to be proven”

6.       I am not sure why Virtual Focus Groups is capitalized. Is this a brand name or commercial product? I think it is a commonly used term and does not need to be capitalized.

7.       Introduction: “according to estimates from the 2017 Eumetra survey, one out of five Italians, or around 10 million citizens, has been 32 treated by an osteopath at least once” – It’s not clear what the Eumetra survey is – perhaps you could briefly describe what this survey is. For example, “According to estimates from a broad survey of the Italian population” or “According to estimates from an Italian population survey…” or “independent research institute”. Also please say “have been treated” rather than “has been treated”

8.       Introduction: “In order to prevent alterations of the musculoskeletal system, the profile's areas of activity and expertise include” – I was not clear what this sentence is saying. Is this simply a general description of osteopathy? Or are you referring to the profile of the Italian Osteopath experts that you studied?

9.       Palpatory Findings does not need to be capitalized

10.   Evidence Based Practice does not need to be capitalized

11.   approximately one every two months – change to “once every”

12.   “This study was submitted to the Institutional Review Board of SOMA Istituto Osteopatia Milano of Milan and approved for informed consent from research participants on data protection and privacy as required by the European Union General Data Protection Regulation.” – Maybe say “of Milan given informed consent was obtained”? (not clear if informed consent was required or waived).  You should also say “This study was approved by” instead of saying “was submitted to”

13.   “A big debate on touch and manual reliability proved that the topic is yet to find full consent throughout the profession with different points of view emerging” – please change “A big debate on” to “Differing viewpoints regarding” or “Divergent view” and change “proved” to “suggested” and “consent” to “agreement”.

14.   Line 260 – I’m not sure if “Anglosassone” is an accurate translation. This sounds like Anglo-Saxon. Should it simply say British or English?

15.   Results: “Assessing tissue quality can play a role in the patient's body awareness and sensitisation” – This section seemed more like a discussion point than a result. Maybe shorten this paragraph and make it from the perspective of the participants?

16.   You could delete the subheading “Strengths, limitations and proposals for the future” in the Discussion.

17.   Line 616: In one of the limitations you mention “friendships” between the participants, but maybe this could be described as “professional relationships” – unless they were all truly friends.

Author Response

Response to Reviewer 2 Comments

Manuscript number healthcare-1844832, entitled "Beliefs and use of palpatory findings in osteopathic clinical practice: a qualitative descriptive study among Italian osteopaths."

We greatly appreciate your readiness to have read our paper and to provide us with relevant feedback and useful suggestions to further improve the quality of our paper. A detailed description of all changes has been provided below.

Major comments

1) Length – I found the manuscript long at almost 30 pages. The introduction seemed to be a good length, as did the Methods. However, much of the excessive length seems to come from the quotes or excerpts. These might be abbreviated further. Also, in some sections you provided 3 quotes, in others you provided four or five. Maybe only 2 or 3 would be needed for each concept. Another idea would be to provide 2 or 3 quotes/examples then include the remainder as a supplemental file. However you approach this, I think you could trim these down where there is redundancy. Also, the Discussion is well written and contains a lot of good information but could be a bit condensed as it is over 5 pages. The conclusions should also be edited to a single brief paragraph.

Response: Thank you for your comments.  We left only two meaningful quotes as examples and added an additional file (Quotes S1) to include those most useful for the analysis done. The discussion was shortened. We would leave the conclusions as they are since they have to summarise different concepts as an outcome of the results obtained from the analysis done and the interpretations, integrations, implications and contributions that emerged from the discussion. After revision the pages became 22 from the 28 they were.

2) Results – In some of the results you provide a brief synopsis of whether there was either agreement or disagreement of the key concept or discussion/question/topic. In other sections, you leave it to the reader to sort through the quotes and determine for themselves what the main finding was. For example, in 3.2.1.3.1. Palpatory Findings, you state “The following quotes present the major opinions emerged…” I would prefer if you made sure each section provided a very brief summary, even 1 sentence, describing the key finding, or even if there is no clear consensus, that would be good to know. I think readers should be able to see the results quickly without needing to rely on reading the quotes.

Response: Thank you for your comments.  We have improved the given example by reviewing the entire sentence (lines 529-548): “Considering what mentioned above, osteopaths engage a relationship with the patient mainly throughout palpation. From this touch-mediated relationship emerge findings that are characteristic of the variables detectable by palpation, such as texture and movement of the tissues:” All other quotes are preceded by a brief summary of the thinking emerging from the panel of experts, where disagreements were present was reported, e.g. lines 303-304.

Minor comments

1) I think the introduction needs 1 sentence providing background on Osteopathy for those not familiar with it. For example, what type of health care provider an osteopath is or just a basic description (e.g., portal of entry, specialist, primary care, allied health practitioner), and what type of conditions they treat.

Response: Thank you for your comments.  We have inserted the following sentence at the beginning of the Introduction part (lines 30-34): “Osteopathy is a ‘person-centered’ approach to the prevention, diagnosis and treatment of illness and injury. Osteopaths are primary-contact health-care professionals that use a range of techniques including ‘hands-on’ manual techniques for assessment and diagnosis to treat various health conditions, as well as musculoskeletal structural problems that influence the body’s physiology.”

2) Maybe somewhere in the methods you should state if the interviews / focus groups were conducted in Italian then translated by the authors (e.g., your initials) into English? (if this is true).

Response: Thank you for your comments.  We inserted the following sentence in the Materials and Methods part (lines 186-188): “The VFGs were conducted in Italian and the related discussions were transcribed into Italian and later translated by the authors into English in order to include them in the quotes identified and reported in the results.”

3) The supplemental file was a little confusing. The figures did not have any axis labels (# of participants, for example?). I’m not sure why the X-axis had 0.5 intervals if these were the number of respondents. Could you have 2.5 or 3.5? It seems all the numbers were whole numbers. Further, the bars could be sorted in order from greatest to least. Otherwise, it is difficult to interpret the figures.

Response: Thank you for your comments. We inserted the following sentence at the beginning of the survey (Survey S1) to specify its meaning and to report the number of participants who responded: “Initial anonymous survey administered to participants (n=12) prior to the start of the VFGs, the results of which gave an idea of the range of opinions in the group regarding PFs in osteopathic clinical practice and to initiate discussion in the fourth VFG”. Labels for the axes have been inserted in Figures 5 and 6 (Participants and 5-point numeric scale: 1 = few, 5 = a lot). The bars for figures 2 and 3 follow the order given in the survey and being multiple choice questions, as stated in the text, the number of answers and the relative percentages given by the participants are shown.

Comments for clarity

1) Abstract: “The Italian government has started the regulatory process for osteopathy to include it among the healthcare professions mentioning terms such as “perceptual palpation” and “somatic dysfunction” within the professional profile” – I was not clear what this means. Does this mean that that government will only regulate osteopathy as a profession provided that these terms are included within the definition of the profession? It is not clear what the government is trying to do to the profession with regards to this terminology. Based on reading this, I’m not sure why the government would care what terms are used by osteopaths. Also, I did not see this type of statement in the Introduction, aside from information stating that the profession is being regulated, so it was difficult to get more clarity on this.

Response: It means that Italian government has started the regulatory process for osteopathy but has not yet concluded it. The implementation of the law involves three steps, the first being the recognition of the professional profile, the second the definition of the university system and the third the equivalence of previous qualifications. At present, the first step has been regulated, i.e. the professional profile, in which the terms “perceptual palpation” and “somatic dysfunction” are mentioned, as set out in more detail in the Introduction part (lines 38-82). We have revised one sentence in that part to make the reading clearer as follows (lines 42-43): “In order to prevent alterations of the musculoskeletal system, the areas of activity and competence of the professional profile include:”.

2) Abstract: “The thematics emerged were” – change to “Themes that emerged were”

Response: Done (line 17).

3) Abstract: “on the territory” – change to “in this field” or “in this profession”

Response: it was modified with “in Italy” (line 14).

4) Abstract: “The results seem to reflect the history of the profession in Italy, which has evolved very quickly embracing tradition and scientific evidence” – I was a little confused by this statement. It seems that some participants embraced concepts with little supporting evidence, having more of a traditional viewpoint, in contrast to others which favored more recent evidence. Tradition and scientific evidence can be opposite things at times, so it seems too broad to say the profession is embracing both things. If the profession is evolving, to me that would imply traditional concepts may be left behind or abandoned in place of newer theories or research. Instead, it seems a little more complicated than that.

Response: Thank you for your comments. That sentence in the abstract summarises concepts that are more explicit in the Discussions and Conclusions parts.

Lines 862-871: “Osteopathy is a very young healthcare profession worldwide compared to medicine and even so in Italy where the first generation of professional osteopaths is still in practice, teaching and at head of OEIs [37]. Tradition with an emphasis on the role of the osteopath and their “listening, seeing hands” [38] is still one of the main features that characterise the way many professionals feel about themselves [39]. In confirmation of this, although Italian osteopaths are in favour of EBP, they lack basic skills in EBP and rarely engage in EBP activities [40] thus maintaining above all a hands-on operator-dependent clinical approach [41], which is supported by some evidence [20-22] but at the same time has shown scientific fragilities in terms of reliability and validity considering the complexity of the phenomenon involved [7, 42-47].”

Lines 1092-1123: “The osteopathic care on the one side includes touch-mediated assessment and treatment for a tissue-mediated relationship with the patient and non-verbal feedback; in this context PFs are considered as diagnostic elements with which the osteopath interacts through a bottom-up approach. On the other side, the hands-off osteopathic approach involves patient management procedures through effective verbal communication for agreement with the patient, such as therapeutic education, health-related advices and self-management strategies for health promotion and prevention. In this case, PFs are used as indicators or moderators of the clinical process in decision-making shared with the patient, giving more importance to the top-down approach. However, as Bohlen suggests [78], both hands-on and hands-off approaches end up involving top-down and bottom-up dynamics. The complexity of the clinical practice often considers these two sides as integrated by EBP, which comprehends the osteopath experience (knowledge, judgement and critical reasoning), the patient preferences (personal and cultural circumstances, values, priorities and expectations) and the best evidence available (external and internal). EBP and the best available evidence play a fulcrum role in the balance between the osteopath's distinctive manual skills and the osteopath's skills for effective communication with the patient, allowing for person-centred care. Education cannot fail to take this balance into account, where tradition, professional distinctiveness, and a biopsychosocial strategy for approaching the person with an informed, EBP will allow professional development in a healthcare context.

Line 1157-1161: “Furthermore, it appears that professional identity is facing a transitional phase in which one looks to the future not yet sure what to leave behind one's past. In this landscape that seems to show uncertainty, the profession has a great opportunity as tradition and evidence coexist. The ability to maintain tradition in a context of informed, evidence-based practice could represent the innovation of osteopathic professional identity.”

To make the sentence more comprehensible to the reader, we have modified the sentence as follows (lines 20-22): “The results seem to reflect the history of the profession in Italy, which has evolved quickly leading professionals to seek new paradigms blending tradition and scientific evidence.”

5) Abstract: “The authors suggest further investigation to verify the state of art among professionals not involved in research or a broader consensus of the results to be proven.” – Maybe change “professionals” to “osteopaths”, and delete “to be proven”

Response: Done (lines 22-24).

6) I am not sure why Virtual Focus Groups is capitalized. Is this a brand name or commercial product? I think it is a commonly used term and does not need to be capitalized.

Response: Thank you for your comments. Virtual Focus Groups has been rewritten with initial letters in lower case.

7) Introduction: “according to estimates from the 2017 Eumetra survey, one out of five Italians, or around 10 million citizens, has been 32 treated by an osteopath at least once” – It’s not clear what the Eumetra survey is – perhaps you could briefly describe what this survey is. For example, “According to estimates from a broad survey of the Italian population” or “According to estimates from an Italian population survey…” or “independent research institute”. Also please say “have been treated” rather than “has been treated”

Response: Thank you for your comments. The sentence was modified as follows (lines 36-38): “Indeed, according to estimates from an Italian population survey carried out in 2017, one out of five Italians, or around 10 million citizens, have been treated by an osteopath at least once.” Furthermore, “has” was replaced with “have”.

8) Introduction: “In order to prevent alterations of the musculoskeletal system, the profile's areas of activity and expertise include” – I was not clear what this sentence is saying. Is this simply a general description of osteopathy? Or are you referring to the profile of the Italian Osteopath experts that you studied?

Response: As stated in the previous sentence (lines 38-41) and as explained above, only the prefessional profile of the osteopath has been recognised at the moment in Italy. Among the areas of activity and competence in the professional profile are the four points mentioned between lines 42-82. As mentioned above, we have revised one sentence in that part to make the reading clearer as follows (lines 42-43): “In order to prevent alterations of the musculoskeletal system, the areas of activity and competence of the professional profile include:”

9) Palpatory Findings does not need to be capitalized

Response: Palpatory Findings has been rewritten with initial letters in lower case.

10) Evidence Based Practice does not need to be capitalized

Response: Evidence Based Practice has been rewritten with initial letters in lower case.

11) approximately one every two months – change to “once every”

Response: Done (line 147).

12) “This study was submitted to the Institutional Review Board of SOMA Istituto Osteopatia Milano of Milan and approved for informed consent from research participants on data protection and privacy as required by the European Union General Data Protection Regulation.” – Maybe say “of Milan given informed consent was obtained”? (not clear if informed consent was required or waived).  You should also say “This study was approved by” instead of saying “was submitted to”

Response: Thank you for your comments. The sentence was modified as follows (lines 149-151): “This study was approved by the Institutional Review Board of SOMA Istituto Osteopatia Milano of Milan for informed consent from research participants on data protection and privacy as required by the European Union General Data Protection Regulation.”

13) “A big debate on touch and manual reliability proved that the topic is yet to find full consent throughout the profession with different points of view emerging” – please change “A big debate on” to “Differing viewpoints regarding” or “Divergent view” and change “proved” to “suggested” and “consent” to “agreement”.

Response: Thank you for your comments. The sentence was modified as follows (lines 291-293): “Differing viewpoints regarding touch and manual reliability suggested that the topic is yet to find full agreement throughout the profession with different points of view emerging.”

14) Line 260 – I’m not sure if “Anglosassone” is an accurate translation. This sounds like Anglo-Saxon. Should it simply say British or English?

Response: Thank you for your comments. “Anglosassone” was changed to “British” (lines 352/371).

15) Results: “Assessing tissue quality can play a role in the patient's body awareness and sensitisation” – This section seemed more like a discussion point than a result. Maybe shorten this paragraph and make it from the perspective of the participants?

Response: Thank you for your comments. We removed the following sentences with their quotes since the reported concepts are discussed in the Discussion part.

Lines 373-378 (previous manuscript): “Assessing tissue quality can play a role in the patient's body awareness and sensitisation. Tissue texture and tenderness/sensitiveness can highlight the patient's comparative signs, which unlike familiar symptoms, can amplify the awareness of the disorder in the patient. The patient's perception with respect to the applied tissue stimulation can also direct the osteopath to distinguish body areas related to the patient's sensitisation. This perception by the patient is shared with verbal feedback, constituting a decision maker.”

Lines 383-385 (previous manuscript): “The quality of the movement expresses the multiplicity of motor choices available and therefore the possibilities of self-regulation and the body's ability to adapt to the demands imposed by the external environment.”

16) You could delete the subheading “Strengths, limitations and proposals for the future” in the Discussion.

Response: Done.

17) Line 616: In one of the limitations you mention “friendships” between the participants, but maybe this could be described as “professional relationships” – unless they were all truly friends.

Response: Thank you for your comments. The sentence was modified as follows (lines 1132-1134): “One factor that may have influenced the limitations of the study is the professional relationships between the participants, which may have hindered the emergence of different points of view, divergences and contrasts.”

Round 2

Reviewer 2 Report

Great job in making revisions and responding to the suggestions. I look forward to seeing the final manuscript. I have no further comments.